# Perinatal Fluoxetine Exposure Has No Major Effect on Myelin-Associated Glycoprotein and Myelin Basic Protein Levels in Auditory Brain Regions

**DOI:** 10.3390/biology14111482

**Published:** 2025-10-24

**Authors:** Joëlle D. Jagersma, Marije Visser, Sonja J. Pyott, Eelke M.S. Snoeren, Jocelien D.A. Olivier

**Affiliations:** 1Department of Otorhinolaryngology and Head/Neck Surgery, University of Groningen, University Medical Center Groningen, 9713 GZ Groningen, The Netherlands; j.d.jagersma@umcg.nl (J.D.J.); s.pyott@umcg.nl (S.J.P.); 2Neurobiology, Groningen Institute for Evolutionary Life Sciences, University of Groningen, 9747 AG Groningen, The Netherlands; 3Department of Neuroscience, School of Medicine and Health Science, Carl von Ossietzky Universität Oldenburg, 26129 Oldenburg, Germany; 4Department of Psychology, UiT The Arctic University of Norway, 9037 Tromsø, Norway; eelke.snoeren@uit.no

**Keywords:** myelination, auditory cortex, fluoxetine, SSRI

## Abstract

Hearing is important for social interactions and learning, but both hearing loss and changes in brain chemistry can affect how the brain develops. One important brain chemical is serotonin, which helps regulate mood and development. Some medications that increase serotonin, such as fluoxetine (commonly prescribed as an antidepressant), are sometimes taken during pregnancy, but it is not clear how they might influence brain development related to hearing. In this study, we tested whether exposure to fluoxetine during pregnancy and early life affected how the brain produces myelin, a protective coating that helps nerve cells transmit signals efficiently. We studied young rats at two different ages and examined brain areas important for hearing and communication. We found that fluoxetine had no major effect on myelin development. However, we did observe changes in myelin as the brain matured, and these changes were different across brain regions. These findings show that brain development in hearing pathways is highly dynamic during early life, but exposure to fluoxetine alone may not disrupt this process. Understanding how medications and hearing-related brain development interact is important for making informed decisions about maternal health and child development.

## 1. Introduction

Hearing loss has been shown to affect social and cognitive behaviors, which potentially contribute to social isolation and cognitive decline [1,2]. However, the underlying neurobiological mechanisms remain poorly understood. A review by Keesom & Hurley [3] suggested that the serotonergic system may be worth exploring in the context of hearing loss, given that dysfunction in both serotonergic signaling and auditory processing can result in similar social and cognitive behavioral outcomes. Serotonin is a key modulatory neurotransmitter involved in a wide variety of behavioral, cognitive and emotional processes, making it a central target in the treatment of various psychiatric and neurological disorders [4]. The serotonin transporter (SERT) plays a crucial role in regulating serotonergic signaling by reabsorbing serotonin from the synaptic cleft back into the presynaptic neuron. SERT is also the primary pharmacological target of selective serotonin reuptake inhibitors (SSRIs), which are commonly used for the treatment of affective disorders [5]. While SERT is known to mediate anxiety- and depression-related behaviors, it may also positively influence auditory processing in older adults with hearing impairments [6]. However, the effects of elevated serotonin levels during development within the context of hearing research remains underexplored, and may differ from the beneficial outcomes observed in adulthood.

Lifelong elevated serotonin levels, such as those observed in SERT knockout (KO) mice, have been associated with notable structural and functional changes in auditory-related brain regions. For instance, SERT KO mice exhibit a reduction in both the number and density of layer IV pyramidal neurons and layer II/III interneurons within the auditory cortex [7]. These mice also demonstrate altered auditory processing, characterized by broader frequency tuning and high-intensity thresholds in auditory neurons. These findings suggest that developmental alterations in the serotonergic system can affect both the neuronal morphology and the functional maintenance of auditory cortical neurons [7]. In a previous study, both mice treated (in adulthood) with the SSRI citalopram (10 mg/kg) as well as SERT KO mice exhibit a reduced mismatch negativity (MMN) response, a cognitive component of auditory event-related potentials that reflects the brain’s ability to discriminate auditory information, indicating deficits in auditory processing [8]. However, this impairment does not appear to involve peripheral hearing function, as auditory brainstem response (ABR) recordings did not reveal any abnormalities [8].

Moreover, serotonin has been shown to affect auditory processes when altered in the developmental period. For instance, exposure of female rats to the serotonergic agonist 5-methoxytryptamine (1 mg/kg) from gestational day (GD)12 to postnatal day (PND)20 resulted in elevated 5-HT levels, and offspring of these rats exhibited heightened reactivity to auditory stimuli, as demonstrated by reduced activity in response to an auditory tone in the open field test [9]. Additionally, rats subjected to in utero undernourishment, which led to elevated levels of L-tryptophan and serotonin in the auditory cortex, showed decreased intensity dependence of the auditory evoked N1/P2 component [10]. These findings suggest that increased developmental serotonin levels in the auditory cortex are associated with altered auditory processing later in life.

The interaction between the serotonergic system and auditory processing seems bidirectional: alterations in hearing can also influence serotonergic functioning. For instance, in a rat model, noise-induced hearing loss was found to reduce serotonergic fiber density and SERT expression in several auditory brain regions, including the auditory cortex and the inferior colliculus [11]. Interestingly, these alterations were partially reversed by treatment with citalopram (20 mg/kg) [11]. Although citalopram did not prevent the hearing damage caused by the noising procedure, it appeared to mitigate the negative impact of hearing loss on the serotonergic system [11].

Multiple findings, both from our own group and others, demonstrate the effects of developmental changes in serotonin on brain alterations. We employed ribonucleic acid (RNA) sequencing to examine the transcriptomic changes caused by (pre- and postnatal) perinatal fluoxetine (10 mg/kg) exposure in the prefrontal cortex and amygdala of 21-day-old rat pups. This analysis revealed significant dysregulation of genes associated with myelination, suggesting that SSRI exposure may alter brain maturation at the molecular level [12]. However, these alterations were measured in the prefrontal cortex and basolateral amygdala and not in auditory-related brain areas. Nevertheless, the finding that perinatal fluoxetine exposure affects myelin-related gene expression is supported by Kroeze et al. [13], who administered fluoxetine (12 mg/kg) to female rats from PND1-PND21 and observed downregulation of myelin-associated genes, including ciliary neurotrophic factor and transferrin, in the adult hippocampus. In a related study, Kroeze et al. [14] examined gene expression in the medial prefrontal cortex of male SERT knockout rats and found that CLDN11, MAG, and MBP expression fluctuated, decreased at PND8, increased at PND14, and decreased again at PND21. Interestingly, these findings contrast with results by Ramsteijn et al. [12], who reported increased myelin-related gene expression in the prefrontal cortex following fluoxetine exposure at PND21. This discrepancy suggests that the effects of developmental SSRI exposure on myelination are highly region- and age-specific and may disrupt circuit synchrony.

In a separate study, we examined the effect of early-onset slight and hidden hearing loss on the corpus callosum and found reduced fluorescent intensity of myelin basic protein (MBP), indicative of impaired myelination [15]. In addition to this, Simpson et al. [16] treated rats with citalopram from either G11-G19 (20 mg/kg), PND1-PND7 (10 mg/kg) or PND8-PND21 (20 mg/kg), and showed that perinatal exposure to citalopram altered the myelination of callosal axons and interfered with oligodendrocytes soma morphology in the corpus callosum. Both hypo- and hypermyelination, together with lamellae separation, was found in the corpus callosum of citalopram-exposed animals. Up to a six-fold increase in aberrant axon morphology was found due to the SSRI exposure, with more robust effects found in postnatal compared to prenatal treatment. Of interest is the fact that Simpson et al. [16] also conducted electrophysiological experiments to test the activity of cortical neurons in the auditory cortex of PND22-PND27 old rats that were exposed to citalopram (5 mg/kg) from PND8-PND21. They showed a distorted tonotopic organization in citalopram-exposed offspring. These findings suggest that SSRI exposure disrupts intracortical network organization, critical for shaping receptive fields and frequency representation during development, which may lead to abnormal auditory processing. Together, these studies raise the question whether other SSRIs also influence myelination in brain areas involved in auditory processing, and whether these effects are detected in myelin-associated proteins across different stages of development.

Given the bidirectional interaction between the serotonergic and auditory systems, and the independent influence of each on neurobiological myelination, this study aims to investigate the effect of perinatal fluoxetine exposure on myelination within (auditory) brain structures (auditory cortex, inferior colliculus, and the corpus callosum). Specifically, we examine the expression of two key markers of myelination: myelin-associated glycoprotein (MAG), which is important for signaling and maintenance of myelin, and MBP, which is essential for the formation and compaction of the myelin sheath [17,18]. We quantified MAG-positive (MAG+) cells counts in the auditory cortex and the inferior colliculus, and assessed MBP expression through measurements of MBP-positive (MBP+) area and fluorescent intensity in the auditory cortex and corpus callosum of rats at PND 21 and 35, as previous work in our laboratory showed that perinatal fluoxetine exposure influences either myelin-associated gene expression or social play behavior at these timepoints [12,19].

## 2. Materials and Methods

### 2.1. Animals and Dam Housing Conditions

A total of 10 female and 10 male Wistar rats (weighing 200–250 g at the time of arrival) were obtained from Charles River (Sulzfeld, Germany) for breeding. The animals were housed in same sex pairs in Makrolon^®^ IV cages (Tecniplast, Buguggiate, Italy) in a room with controlled temperature (21 ± 1 °C) and humidity (55 ± 10%) on a 12:12 h light/dark cycle (lights on 11:00 h). Commercial rat pellets (Standard chow from SDS, Special Diet Services) and tap water were provided ad libitum, and nesting material was present. All experimentation was carried out in agreement with the European Union council directive 2010/63/EU. The protocol was approved by the National Animal Research Authority in Norway (#22453) on the 17 April 2020.

### 2.2. Breeding and Antidepressant Treatment

Prior to breeding, females were checked daily for their estrus cycle stage by placing them together with a male rat for a maximum 5 min. They were considered receptive when they responded to a mount with a lordosis response. When receptive, the females were housed with a male for approximately 24 h, at GD0. After 24 h, both male and female returned to their original home cage (with same-sex partner) for the first two weeks of pregnancy. On GD14, the females were housed singly with access to nesting material.

From GD1 until PND21, females were randomly assigned to either the experimental group, receiving daily 10 mg/kg fluoxetine (Apotekproduksjon, Oslo, Norway), or the control group, receiving vehicle (1% methylcellulose; Sigma-Aldrich, St. Louis, MO, USA), via gavage using a stainless-steel feeding needle (total of 6 weeks). In our previous study, we found that maternal oral administration of 12 mg/kg fluoxetine resulted in a plasma transfer of 83%, with fluoxetine levels in the pup brain reaching levels of up to 13.0 µg/g [20]. Fluoxetine tablets (for human usage) were pulverized and dissolved in sterile water (2 mg/mL) and injected at a volume of 5 mL/kg. As a control condition, methylcellulose, the non-active filling of fluoxetine tablets, was dissolved in sterile water to create a 1% solution and administered at a volume of 5 mL/kg as well. The amount of vehicle/fluoxetine given was adjusted upon the weight of the females who were weighed every three days. Near the end of the pregnancy, dams were checked twice a day (9:00 h and 15:00 h) for pup delivery. From PND21 pups were weaned and were housed with littermates until PND35.

### 2.3. Brain Collection

Perfusion of pups took place at either PND21 or PND35. To reduce the total number of animals used in research, this study utilized surplus male pups from a related study. For PND21, 7 control and 6 fluoxetine-treated male rats were used, and for PND35 6 control and 8 fluoxetine-treated male rats were used. Rats were injected intraperitoneally with 2 mL/kg ZRF-mixture (2.9 mg Zolazepam, 12.9 mg Tiletamine, 1.8 mg Xylazine, and 10.3 μg Fentanyl per mL) and subsequently perfused transcardially with phosphate-buffered saline (PBS) followed by 4% formaldehyde. Brains were removed and postfixed overnight at 4 °C in 4% formaldehyde. The tissue was then rinsed and cryoprotected in (sequentially) 10%, 20%, and 30% sucrose in PBS + 0.1% proClin (Sigma-Aldrich, St. Louis, MO, USA) and consequently, they were shipped to the Olivier lab at the University of Groningen and stored at 4 °C for until further used.

### 2.4. Tissue Sectioning

After storage, the terminal end of the spinal cord was removed, and the brains were rapidly frozen using liquid nitrogen. The brains were then stored at −70 °C, until further processing. Prior to sectioning, brains were transferred to a −20 °C freezer, for a period ranging from 18 h to just over 72 h. Sectioning was performed using a sliding microtome (HM 450, MICROM International GmbH, Walldorf, Germany). Brains were cut into 40 μm thick coronal sections and stored in 0.01 M PBS and 0.1% sodium azide at 4 °C.

### 2.5. Immunohistochemistry

Brain sections were washed 3 times for 5 min in 0.01 M PBS, after which they were submerged in a blocking solution (0.01 M PBS, 5% BSA, 0.2% Triton X-100 (Sigma-Aldrich, St. Louis, MO, USA) for two hours at room temperature. Subsequently, sections were incubated in anti-MAG (Abcam, Cambridge, UK, ab277524, 1:10.000) and anti-MBP (Biolegend, United states, San Diego, CA, USA, #808401, 1:500) primary antibodies overnight at 4 °C. The following day, the sections were washed 3 times for 5 min in 0.01 M PBS and consequently incubated for two hours with the secondary antibodies (donkey anti-mouse, Invitrogen, Thermo Fisher Scientific, United States, Grand Island, NY, Alexa fluor 488, 1:500 and goat anti-rabbit, Invitrogen, Alexa fluor 555, 1:500) at room temperature. Sections were mounted using ProLong Gold Antifade (with DAPI) (Thermo Fisher Scientific, United States, Grand Island, NY, USA, P36931).

### 2.6. Microscopy and Image Analysis

The investigated regions of interest included the auditory cortex (bregma: −4.68 to −6.36 mm), inferior colliculus (bregma: −8.16 to −9.12), and corpus callosum (bregma: 3.72 to −3.24 mm), and Appendix A includes results from the medial prefrontal cortex (bregma: 3.72 to 2.52) and basolateral amygdala (bregma: −1.92 to −3.24). For each analysis, 4 to 6 brain sections were measured per region of interest, and were averaged per experimental animal during analysis. Experimenters were blinded to treatment allocation during outcome assessment.

For visualizing anti-MAG immunofluorescence, a Zeiss AxioObserver Z1 Tissuefaxs (Carl Zeiss AG, Germany, Jena), was used to image sections containing the regions of interest at a magnification of 10×. The MAG immunoreactivity was analyzed by both manual and semi-automatic counting of MAG+ cells using FIJI (version 2.9.0) [21]. Initially, a threshold was established through visual assessment to remove background noise. The cell counter plugin was used for the cell counting in the prefrontal cortex and amygdala, and the analyze particle function of FIJI was used to quantify MAG+ cells in the auditory cortex and inferior colliculus. Additionally, the area of the regions of interest were measured, resulting ultimately in the measure of the number of MAG+ cells per mm^2^.

For visualizing anti-MBP immunofluorescence, a Leica SP8 (Leica Microsystems, Germany, Wetzlar) confocal microscope was used to image sections containing the regions of interest at a magnification of 40×. Different measures of MBP immunoreactivity were assessed. First, the spatial extent of brain myelination was quantified by measuring the MBP+ area. A manual threshold was applied to capture all stained regions, and the proportion of the image field occupied by MBP signal was calculated. Second, the corrected fluorescence intensity was used to estimate the amount of MBP-related signal, expressed in arbitrary units. It was calculated as: the integrated density—(area of the region of interest × mean background fluorescence), following the approach of Bora et al. [22]. By combining measurements of MBP-positive area and fluorescence intensity, it is possible to distinguish between different developmental or treatment-related effects. For example, a brain region may exhibit increased myelination efficiency (higher intensity) despite a reduction in overall myelinated area, potentially reflecting processes such as refinement or pruning.

The main text reports MAG+ cell counts in the auditory cortex and inferior colliculus, while MBP analyses were conducted in the auditory cortex and corpus callosum. The inferior colliculus was excluded from MBP measurements due to anatomical limitations: in coronal sections, many of its fiber tracts are cut transversely, making accurate quantification of MBP signal difficult [23]. Instead of the inferior colliculus, the corpus callosum was included for MBP measurements, based on prior findings from our lab showing that early-onset hearing loss reduces MBP fluorescence intensity in this region [15]. Additional results from the prefrontal cortex and basolateral amygdala are provided in Appendix A, as the main focus of this study was on auditory-related brain areas.

### 2.7. Statistics

The collected data from FIJI was statistically analyzed in Prism 10 (GraphPad, La Jolla, CA, USA). Three- or two-way ANOVAs were performed for comparisons in which three or two factors influenced the outcome parameter of interest. The specific factors are described in more detail for each analysis in the results section. Normality of the data was tested using Shapiro–Wilk tests in combination with visual inspection of the data using Q-Q plots. However, in case one of the assumptions of an ANOVA was violated, mixed-effect analyses were performed, where fluoxetine treatment, developmental timepoint and brain region were used as fixed effects, and the individual rats were used as random effects. If the sphericity assumption was violated in any of the analyses, a Greenhouse–Geisser correction was applied. For each analysis, all investigated brain regions were included in the analysis, followed by a Šídák multiple comparison test. Occasional missing values due to dissection or processing artifacts are reflected in the reported sample sizes in the corresponding figures. Data are presented as mean ± standard error. *p* values of less than 0.05 were considered statistically significant for all tests.

## 3. Results

### 3.1. Myelin-Associated Glycoprotein Expression in the Auditory Cortex and Inferior Colliculus

Counts of MAG+ cells were quantified in both the auditory cortex and inferior colliculus of control and fluoxetine-treated rats at PND21 and PND35 (Figure 1). A mixed-effects analysis revealed significant effects of both brain region (F(1.934, 38.68) = 54.05, *p* < 0.0001) and developmental stage (F(1, 24) = 42.80, *p* < 0.0001), while fluoxetine treatment did not significantly influence MAG cell counts (F(1, 24) = 0.6152, *p* = 0.4405). Moreover, an interaction between brain region and developmental stage was observed (F(3, 60) = 16.90, *p* < 0.0001), yet no interactions were found between brain region and fluoxetine treatment (F(3, 60) = 0.2418, *p* = 0.8668), developmental stage and fluoxetine treatment (F(1, 24) = 1.170, *p* = 0.2901) or brain region, developmental stage and fluoxetine treatment (F(3, 60) = 0.3359, *p* = 0.7994). Similarly, no significant effects of fluoxetine on MAG+ cells were observed in the prefrontal cortex or basolateral amygdala (Appendix A).

Given the lack of an effect of fluoxetine, treatment groups were added together to further investigate developmental changes in MAG expression across brain regions. This mixed-effects analysis again indicated a significant effect of brain region (F(1.948, 42.86) = 57.25, *p* < 0.0001), developmental stage (F(1, 26) = 42.62, *p* < 0.0001), and an interaction between these variables (F(3, 66) = 17.68, *p* < 0.0001). A Šídák multiple comparison test revealed a significant effect in MAG+ cells, with more MAG+ cells in the inferior colliculus at PND21 compared to PND35 (*t*(11.99) = 6.105, *p* = 0.0002). In contrast, no developmental changes in MAG expression were observed in the auditory cortex (*t*(22) = 2.102, *p* = 0.1760).

### 3.2. Myelin Basic Protein Expression in Auditory Cortex and Corpus Callosum

First, MBP+ area was quantified in both the corpus callosum and the auditory cortex of control and fluoxetine-treated rats at PND21 and PND35 (Figure 2). A mixed-effects analysis revealed significant effects of both brain region (F(1.881, 37.62) = 136.5, *p* < 0.0001) and developmental stage (F(1, 22) = 134.1, *p* < 0.0001), while fluoxetine treatment did not significantly influence MBP+ area (F(1, 22) = 2.842, *p* = 0.1060). Moreover, an interaction between brain region and developmental stage was observed (F(3, 60) = 67.98, *p* < 0.0001), yet there were no interactions between brain region and fluoxetine treatment (F(3, 60) = 0.8718, *p* = 0.4608), developmental stage and fluoxetine treatment (F(1, 22) = 0.7147, *p* = 0.4070), or brain region, developmental stage and fluoxetine treatment (F(3, 60) = 1.167, *p* = 0.3298). Similarly, no significant effects of fluoxetine on MBP+ area were observed in the prefrontal cortex or basolateral amygdala (Appendix A).

Given the lack of an effect from fluoxetine, treatment groups were added together to further investigate developmental changes in MBP+ areas across brain regions. This mixed-effects analysis again indicated a significant effect of brain region (F(1.885, 41.47) = 136.3, *p* < 0.0001), developmental stage (F(1, 24) = 127.5, *p* < 0.0001), and an interaction between these variables (F(3, 66) = 68.14, *p* < 0.0001). A Šídák multiple comparison test showed a significant reduction in the MBP+ area in PND35 compared to PND21 in the both corpus callosum (*t*(21.03) = 12.05, *p* < 0.0001) and the auditory cortex (*t*(12.06) = 5.006, *p* = 0.0012).

Second, corrected fluorescent intensity of MBP was also quantified in both the corpus callosum and the auditory cortex of control and fluoxetine-treated rats at PND21 and PND35 (Figure 2G,H). A mixed-effects analysis revealed significant effects of both brain region (F(1.399, 27.52) = 44.27, *p* < 0.0001) and developmental stage (F(1, 22) = 7.507, *p* = 0.0120), while fluoxetine treatment did not significantly influence corrected fluorescent intensity (F(1, 22) = 0.3015, *p* = 0.5885). Moreover, an interaction between brain region and developmental stage was observed (F(3, 59) = 7.389, *p* = 0.0003), yet there were no interactions between brain region and fluoxetine treatment (F(3, 59) = 0.1659, *p* = 0.9190), developmental stage and fluoxetine treatment (F(1, 22) = 0.00006, *p* = 0.9939), or brain region, developmental stage and fluoxetine treatment (F(3, 59) = 0.1506, *p* = 0.9289). Similarly, no significant effects of fluoxetine in MBP intensity were observed in the prefrontal cortex or basolateral amygdala (Appendix A).

Given the lack of an effect from fluoxetine, treatment groups were added together to further investigate developmental changes in corrected fluorescent intensity of MBP across brain regions. This mixed-effects analysis indicated a significant effect of brain region (F(1.414, 30.64) = 47.63, *p* < 0.0001), developmental stage (F(1, 24) = 8.666, *p* = 0.0071), and an interaction between these variables (F(3, 65) = 7.815, *p* = 0.0002). A Šídák multiple comparison test showed a significant increase in corrected fluorescent intensity of MBP in the corpus callosum in PND35 compared to PND21 (*t*(22.14) = 3.105, *p* = 0.0204). In contrast, no developmental changes in corrected fluorescent intensity of MBP expression were observed in the auditory cortex (*t*(22) = 0.9132, *p* = 0.8435).

## 4. Discussion

This study investigated the effects of perinatal fluoxetine exposure on myelination of (auditory) brain regions by quantifying expression of myelin-associated glycoprotein (MAG) and myelin basic protein (MBP) in the auditory cortex, inferior colliculus, and corpus callosum at postnatal days (PND) 21 and 35. No major effects of fluoxetine were observed on myelin-associated markers in any investigated brain region or developmental stage, and no interactions were detected between treatment and brain region on MAG+ or MBP+ cell counts. In contrast, significant developmental changes in myelin-associated proteins were observed between PND21 and PND35. MAG+ cell counts decreased over time in the inferior colliculus but remained stable in the auditory cortex. MBP expression showed a reduction in MBP+ area from PND21 to PND35 in both the corpus callosum and auditory cortex, while fluorescent intensity of MBP increased in only the corpus callosum, but remained stable in the auditory cortex. These results suggest that developmental stage and brain region, but not fluoxetine exposure, primarily drove changes in myelin-related markers between PND21 and PND35.

The absence of major fluoxetine-related effects on myelin-associated proteins should be interpreted with caution and within the context of the present study’s design. One potential factor is the absence of additional experimental stress induction. Increased anxiety levels have been documented in both SERT KO mice and rats, and similar increases in anxiety have been found in rats exposed to fluoxetine during pregnancy and lactation [5,20,24,25]. Interestingly, we previously found altered RNA expression of myelin-related genes, including those encoding MAG and MBP, especially when fluoxetine exposure was combined with early life stress induction in dams [12]. In the current study, fluoxetine was administered in the absence of experimental maternal stress induction, suggesting that stress may be a necessary modulator of fluoxetine’s impact on neurodevelopment. It should be noted that the alterations in RNA expression in the Ramsteijn et al. [12] study were found in the prefrontal cortex and basolateral amygdala, and showed opposite expression levels between these brain areas. In the present study, however, we did not observe corresponding changes at the protein level in these regions (see Appendix A) or in auditory-related areas. This finding contrasts with robust findings of both hypo- and hypermyelination after perinatal citalopram exposure, particularly in the corpus callosum, where oligodendrocyte morphology and axonal organization were markedly altered [16]. Together, these findings suggest that SSRI effects on myelination may be compound-specific, as fluoxetine and citalopram differ in their pharmacokinetic [26] and receptor-binding properties [27]. Future research should include ultrastructural studies to assess the architecture of axons in the corpus callosum, inferior colliculus, and auditory cortex to confirm this. At the same time, the divergence between altered myelin-related mRNA expression and stable protein levels across multiple regions underscores the importance of considering molecular level, regional, and developmental specificity when evaluating serotonergic modulation of myelination [13,14]. Moreover, auditory-related brain areas may exhibit both hypo- and hypermyelination, as demonstrated by Simpson et al. [16] in rats perinatally exposed to citalopram. As explained in the introduction, such opposing effects could potentially mask the influence of fluoxetine on MAG and MBP protein expression levels. Additionally, compensatory developmental mechanisms may have mitigated potential fluoxetine-induced disruptions. These mechanisms could include post-transcriptional regulatory processes or shifts in the timing of gene and protein expression that help maintain functional myelin protein levels during sensitive windows of development [14,28]. Because perinatal exposure to citalopram did induce hypo- and hypermyelination in the corpus callosum of rats, and caused distorted tonotopic organization in the auditory cortex [16], this explanation seems less plausible. Finally, it also remains possible that our study was not sufficiently powered to detect subtle protein-level effects of fluoxetine. Nevertheless, our findings refine the picture emerging from prior studies by showing that perinatal fluoxetine exposure does not universally produce major alterations of myelin markers, emphasizing the need to consider drug compound, brain region, developmental stage, and experimental power when reconciling the heterogeneous literature on SSRI-induced changes in myelin.

The observed developmental changes in MAG and MBP expression are consistent with known trajectories of oligodendrocyte maturation and synaptic pruning. MAG is predominantly expressed by immature oligodendrocytes [18,29], and the decline in MAG+ cell counts in the inferior colliculus may reflect a transition toward a more mature oligodendrocyte phenotype. MBP, in contrast, is more strongly expressed by mature oligodendrocytes [30], and the increase in MBP fluorescent intensity in the corpus callosum could suggest that this region started maturing at PND35 compared to PND21. However, a reduction in MBP+ area was observed at PND35 across all investigated regions, possibly reflecting pruning and axonal remodeling processes that reduce the overall area requiring myelination. These processes typically intensify during adolescence, which begins shortly after weaning in rodents [31]. Thus, local increases in MBP fluorescent intensity may occur alongside global reductions in MBP+ area due to developmental refinement of neural circuits.

Not all regions examined in this study exhibited this suggested developmental trajectory in expression of myelin markers, which might be due to the timing of the data collection (PND21 and 35). Myelination does not occur uniformly across the brain during development. Previously it was believed that caudal brain regions generally myelinate earlier, followed by rostral progression during development [32]. However, more recent work shows that the timing of myelination of specific brain regions might also be function-specific, with brain regions that are important during early development (e.g., the suckling reflex) myelinating earlier, whereas brain regions involved in more complex cognitive processes mature later [33].

A few other considerations should be made when interpreting the current findings. First, no assessments of auditory system functionality were performed, making it unclear whether the perinatal exposure to fluoxetine resulted in any functional changes of the auditory system. As a result, it is also uncertain whether potential auditory changes could have consequently contributed to expression of the investigated myelin markers. Second, this study relied on surplus animals from another experiment, which may have resulted in sample sizes that were too small to detect potential fluoxetine effects with smaller effect sizes. Third, previous research in SERT knockout mice revealed layer-specific alterations, including a reduction in dendritic spine density in layer four pyramidal neurons and second-to-third layer interneurons of the auditory cortex [7]. However, our aim was to characterize broad region-level effects of perinatal fluoxetine on cortical development. Resolving laminar differences required a dedicated design and power that was beyond the scope of the present study. Fourth, while MAG expression was quantified based on the number of MAG+ cells, an estimation of total protein levels was not measured, which may obscure more subtle changes in protein abundance or turnover. Complementary molecular approaches, such as Western blotting, (quantitative) PCR, or proteomic analyses, could provide a more comprehensive assessment of both overall protein levels and underlying regulatory mechanisms of the investigated myelin-associated protein. Lastly, by focusing on two developmental timepoints, this study provides a focused snapshot of the myelination process. While the interaction between brain region and developmental trajectory was beyond the scope of this investigation, it is an avenue for future research and should be considered when interpreting the findings.

## 5. Conclusions

In conclusion, the developmental changes observed in MAG and MBP expression within the auditory pathway highlight the dynamic and region-specific nature of myelination during critical periods of postnatal brain development. Proper myelination is essential for the efficient transmission of (auditory) signals, and disruptions during these sensitive windows may have long-lasting effects on auditory processing and related cognitive and social functions. Our findings demonstrate that perinatal fluoxetine exposure does not cause major, large-scale alterations in myelin protein levels within auditory brain regions. However, due to the limited sample size, our study cannot rule out more modest effects that may still hold functional significance. Further research is necessary to evaluate potential long-term consequences and ultrastructural changes, and it should examine later developmental stages and include (behavioral) assessments of auditory processing and social and cognitive functioning. Moreover, layer-specific and cellular changes in oligodendrocyte populations should be investigated. These efforts will be essential to determine whether early fluoxetine exposure exerts subtle, delayed, or context-dependent effects on myelination and neural function.

## Figures and Tables

**Figure 1 biology-14-01482-f001:**
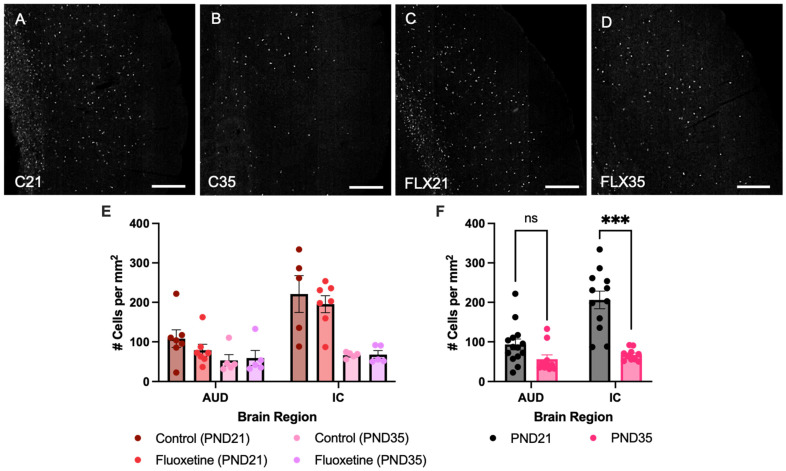
**Myelin-associated glycoprotein positive cells in the auditory cortex and inferior colliculus.** Myelin-associated glycoprotein (MAG)-stained brain sections containing the auditory cortex for control rats at PND21 and PND35 ((**A**,**B**), respectively), and fluoxetine-treated rats at PND21 and PND35 ((**C**,**D**), respectively). Scale bar = 333 μm, C = control, FLX = fluoxetine, PND = postnatal day. MAG+ cell counts are presented for both treatment groups individually (**E**) and for the treatment groups averaged together (**F**). # = number of cells, AUD = auditory cortex, IC = inferior colliculus, ns = not significant, and *** = *p* ≤ 0.001.

**Figure 2 biology-14-01482-f002:**
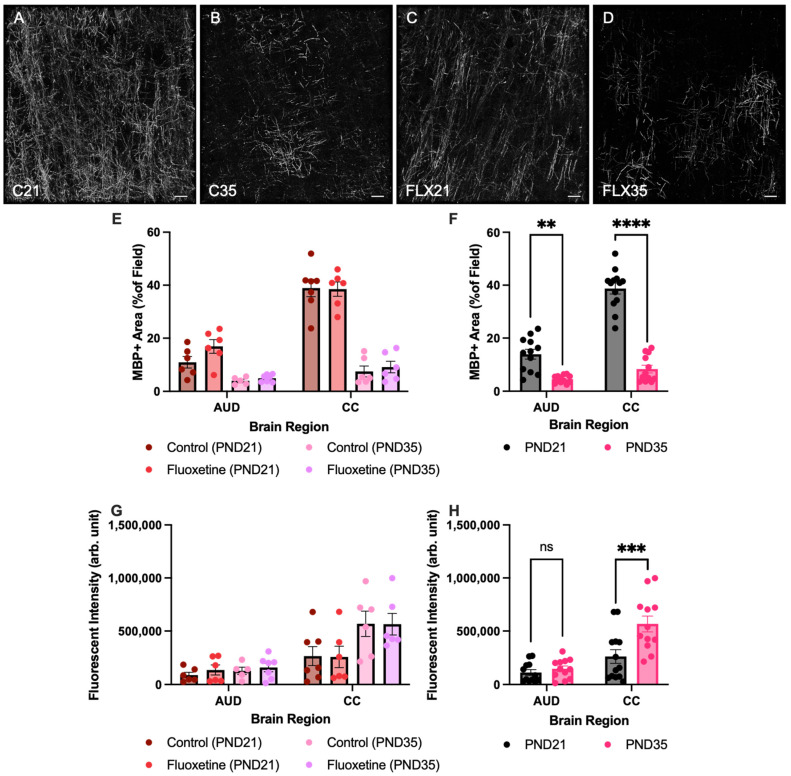
**Myelin basic protein positive area and fluorescent intensity in the auditory cortex and corpus callosum.** Myelin basic protein (MBP)-stained brain sections containing the auditory cortex for control rats at PND21 and PND35 ((**A**,**B**), respectively), and fluoxetine-treated rats at PND21 and PND35 ((**C**,**D**), respectively). Scale bar = 20 μm, C = control, FLX = fluoxetine, PND = postnatal day. MBP+ area is presented for both treatment groups individually (**E**) and for the treatment groups averaged together (**F**). Analysis of fluorescent intensity of MBP was also performed in both treatment groups individually (**G**) and for the treatment groups averaged together (**H**). AUD = auditory cortex, CC = corpus callosum, ns = not significant, ** = *p* ≤ 0.01, *** = *p* ≤ 0.001 and **** = *p* ≤ 0.0001.

## Data Availability

A minimal dataset supporting the conclusions of this study is available in the Appendix A. The raw data generated and analyzed during the current study are available from the corresponding author upon reasonable request.

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
