# Peer review of "Perinatal Fluoxetine Exposure Has No Major Effect on Myelin-Associated Glycoprotein and Myelin Basic Protein Levels in Auditory Brain Regions"

_biology, 2025, doi:10.3390/biology14111482_

Round 1
Reviewer 1 Report
Comments and Suggestions for Authors
The masnucript enetitled "Perinatal fluoxetine exposure does not alter myelin associated glycoprotein and myelin basic protein levels in auditory brain regions" is well written in clear understandable English. Authors analyze the changes in auditory area of the brain of postnatal rats and also analyze the effect of fluoxetine on the mielinization of auditory area. This manuscript is a type of negative-results manusciprts as the authors did not find significant action of fluorexetine on auditory region of postnatal rats. It is very important for scientific community to report also negative results too. However, I think, authors may improve their manuscript. Below I wll provide my recommendations which can be divided into Major and Minor
Major
- You provided for the animals fluorexetin by oral gavage. Are you sure that animals really obtained fluorexetin? Oral gavage is not stable method for delivery of drugs in comparison for example with injections.
- Are you sure that dose of luorexetin which you used is sufficient?
- Are you sure that fluorexetin was addresed to the brain and not metabolized aerlier, for example, in the liver? It will be well to provide some proofs that fluorexetine of treated animals achieve the brain.
- The bumber of Methods which were use din the work is limited. It will be interesting to see not only ICH but also PCR.
Minor
- Line 102. Provide full terms of abbreviations
- Introductions should be shortened and a part where prvious works are described should be moved to the discussion section.
- Statistics. You used ANOVA. How did you checked that your data have normal distribution?
Author Response
Thank you very much for taking the time to review our manuscript. Please find detailed responses to your comments below.
Major
Comment 1: You provided for the animals fluorexetin by oral gavage. Are you sure that animals really obtained fluorexetin? Oral gavage is not stable method for delivery of drugs in comparison for example with injections.
Response 1: When applying fluoxetine to animals via oral gavage, the drug solution is administered directly into the stomach using a stainless steel gavage needle. This means the animals do not voluntarily ingest the drug; instead, it is applied by the researchers in a controlled manner. Oral gavage is a very effective method of administration, and oral application of 10 mg/kg fluoxetine has been shown to reach 2.4 ± 0.5 nmol/g in the brain, with a plasma Tmax of 3.3 ± 1.6 nmol/mL (Caccia et al., 1990). Therefore, we are very confident that the animals did indeed obtain fluoxetine via this method. See also related responses to question 1-3.
Caccia, S., Cappi, M., Fracasso, C., & Garattini, S. (1990). Influence of dose and route of administration on the kinetics of fluoxetine and its metabolite norfluoxetine in the rat. Psychopharmacology, 100(4), 509-514.
Comment 2: Are you sure that dose of fluorexetin which you used is sufficient?
Response 2: Based on our previous experiments, we are confident that the dose of fluoxetine used in these experiments is sufficient. We applied the same approach (administration of 10 mg/kg fluoxetine via oral gavage) when studying gene expression alterations in the offspring brain, where we observed changes in myelin-related genes in the prefrontal cortex and amygdala (Ramsteijn et al., 2022). These findings were based on mRNA expression levels rather than protein expression. Nevertheless, they provide strong evidence that this dose is effective in altering myelin-associated mRNA levels in the brain. This does not rule out that other doses would show other effects, but based on our gene expression data, we are convinced the dose is sufficient. See also related responses to question 1-3.
Ramsteijn, A. S., Verkaik-Schakel, R. N., Houwing, D. J., Plösch, T., & Olivier, J. D. (2022). Perinatal exposure to fluoxetine and maternal adversity affect myelin-related gene expression and epigenetic regulation in the corticolimbic circuit of juvenile rats. Neuropsychopharmacology, 47(9), 1620-1632.
Comment 3: Are you sure that fluorexetin was addresed to the brain and not metabolized aerlier, for example, in the liver? It will be well to provide some proofs that fluorexetine of treated animals achieve the brain.
Response 3: We previously found this method effective for the transfer of fluoxetine from mother to pup, achieving a plasma transfer of 83% and 13.0 µg/g in the pup brain following oral administration of 12 mg/kg to the mothers (Olivier et al., 2011). And as explained under point 2, we did see previous effects of this dose in brain gene expression. We added this to the revised manuscript because this explanation was indeed lacking (line 176-179). The new text reads: “In our previous study, we found that maternal oral administration of 12 mg/kg fluoxetine resulted in a plasma transfer of 83%, with fluoxetine levels in the pup brain reaching levels of up to 13.0 µg/g [20]”. See also related responses to question 1-3.
Olivier, J. D., Vallès, A., van Heesch, F., Afrasiab-Middelman, A., Roelofs, J. J., Jonkers, M., ... & Homberg, J. R. (2011). Fluoxetine administration to pregnant rats increases anxiety-related behavior in the offspring. Psychopharmacology, 217(3), 419-432.
Comment 4: The bumber of Methods which were use din the work is limited. It will be interesting to see not only ICH but also PCR.
Response 4: We agree with the reviewer that the number of methods used in our study is limited. This point has been acknowledged as a limitation in the discussion (line 444-447). The new text reads: “Complementary molecular approaches, such as Western blotting, (quantitative) PCR, or proteomic analyses, could provide a more comprehensive assessment of both overall protein levels and underlying regulatory mechanisms of the investigated myelin associated protein.”. While PCR analysis could indeed provide valuable complementary molecular insights (as we showed previously in the prefrontal cortex and amygdala), it was not feasible within the scope of this study, as it would require additional non-perfused brain material that was not available. For future studies we will definitely include qPCR studies for the auditory brain areas as well.
Minor
Comment 1: Line 102. Provide full terms of abbreviations
Response 1: We included “ribonucleic acid” to introduce the RNA abbreviation in the revised version.
Comment 2: Introductions should be shortened and a part where prvious works are described should be moved to the discussion section.
Response 2: We understand this concern; however, reviewer 2 indicated that the introduction did not yet include enough background and relevant references. Therefore, we decided to leave the mentions of the previous work in the introduction, but we still refer back to this literature in the discussion (line 369-402).
Comment 3: Statistics. You used ANOVA. How did you checked that your data have normal distribution?
Response 3: We tested normality using the Shapiro-Wilk test in combination with visual inspection of the data using Q-Q plots. These details have been added to section 2.7 (line 261-263) in the revised version. The revised text reads: “Normality of the data was tested using Shapiro-Wilk tests in combination with visual inspection of the data using Q-Q plots.”.
Reviewer 2 Report
Comments and Suggestions for Authors
The study presents a well-designed investigation into the effects of perinatal fluoxetine exposure on myelination markers in auditory brain regions. The methodology is sound, the introduction provides a compelling rationale, and the results are clearly presented. The finding that myelination dynamics between PND21 and PND35 are region- and age-specific is a valuable contribution to the field. However, several key points require clarification and additional analysis to strengthen the manuscript and fully support its conclusions.
- The manuscript's central conclusion is the absence of a significant effect of fluoxetine on myelin markers. To robustly support this claim and rule out a Type II error (failing to detect a true effect), it is essential to include a power analysis. Could the authors please provide a post-hoc power calculation based on the observed effect sizes (e.g., Cohen's d or η²) for the fluoxetine treatment factor in their key comparisons? Including these effect sizes and the corresponding power (1-β) in the results section would greatly strengthen the argument that the lack of significance is due to a genuine absence of effect rather than insufficient sample size. Furthermore, the discussion should be tempered to acknowledge that while no statistically significant effect was found with the current sample size, the study may have been powered to detect only larger effects, and more subtle influences of fluoxetine cannot be entirely ruled out.
- The use of mixed-effects models is appropriate for the complex design. However, the reporting could be more complete to ensure reproducibility. Please specify the exact structure of the mixed models used (e.g., the random effects included, such as the intercept for each animal). Additionally, for analyses reporting degrees of freedom with decimals (e.g., F(1.934, 38.68)), please indicate if a specific correction method (e.g., Greenhouse-Geisser) was applied and the rationale for its use (e.g., violation of sphericity). This transparency is crucial for the reader to fully evaluate the statistical approach.
- The discussion section would benefit from a more nuanced interpretation of the negative findings. While the authors rightly mention potential compensatory mechanisms and the absence of maternal stress as mitigating factors, the language could more explicitly frame the results within the context of the study's limitations, particularly the statistical power point raised above. Acknowledging that the findings demonstrate a lack of a major effect of perinatal fluoxetine on these specific myelin proteins at these specific timepoints would provide a more precise and defensible conclusion. This would nicely set the stage for the excellent suggestions for future research that the authors already provide.
- The introduction effectively outlines the contradictory literature on SSRIs and myelination. The novelty of the current study—focusing on auditory regions and a specific SSRI (fluoxetine) at key developmental stages—could be more sharply articulated in the discussion. Please expand on how your findings help reconcile the discrepancies between previous studies. Does the absence of an effect with fluoxetine suggest that the robust effects seen with citalopram are compound-specific? Does it highlight the critical importance of brain region and timing? A more direct discussion positioning your null result within the existing conflicting literature would significantly enhance the impact of the manuscript.
- In the Methods section (2.6), it is stated that the mPFC and BLA results are in Supplementary Material 1. For completeness, it would be helpful to briefly summarize the key outcome of the fluoxetine treatment in these regions in the main text results (e.g., "Similarly, no significant effects of fluoxetine were observed in the mPFC or BLA; see Supplementary Material 1")
Author Response
Thank you very much for taking the time to review our manuscript. Please find detailed responses to your comments below.
Comment 1: The manuscript's central conclusion is the absence of a significant effect of fluoxetine on myelin markers. To robustly support this claim and rule out a Type II error (failing to detect a true effect), it is essential to include a power analysis. Could the authors please provide a post-hoc power calculation based on the observed effect sizes (e.g., Cohen's d or η²) for the fluoxetine treatment factor in their key comparisons? Including these effect sizes and the corresponding power (1-β) in the results section would greatly strengthen the argument that the lack of significance is due to a genuine absence of effect rather than insufficient sample size. Furthermore, the discussion should be tempered to acknowledge that while no statistically significant effect was found with the current sample size, the study may have been powered to detect only larger effects, and more subtle influences of fluoxetine cannot be entirely ruled out.
Response 1: We appreciate the Reviewer's emphasis on rigorously supporting our conclusion regarding the absence of a significant fluoxetine effect and avoiding a Type II error. We agree that maximum statistical transparency is essential.However, we decided to not include a post hoc power analysis as there has been evidence that shows such analyses can be uninformative and potentially misleading (Heinsberg & Weeks, 2023). Using observed power to argue a study was "underpowered" is circular logic and encourages misinterpretation, since it confuses the lack of evidence with the evidence of a lack of effect.
We believe the most transparent approach for our reporting is to provide the raw information, allowing readers to conduct their own prospective power calculations based on a meaningful effect size. Therefore, the manuscript reports F-statistics and the degrees of freedom for all analyses, which can be used to calculate effect size (partial eta-squared). Together with the additional descriptive statistics readers can perform their own power analyses if needed.
Moreover, the current study was conducted using surplus animals from a related experiment, which we have clarified in the source of animals in the Methods (line 189–190). Consequently, an a priori sample size calculation was not performed. However, based on previous a priori power analyses performed in our laboratory for comparable research (using an effect size of 0.4, an alpha level of 0.05, and a power of 0.8) a sample size of 6.8 rats was determined to be sufficient to minimize the risk of type II error. In the present study, the sample size in some analyses was indeed at the lower end of this threshold. For this reason, we have nuanced our discussion and explicitly acknowledged that potential fluoxetine effects of smaller magnitude may not have been detected due to the limited sample size, and we have addressed the implications of this limitation for the interpretation of our findings (line 402-403 & 434–437).
Heinsberg, L. W., & Weeks, D. E. (2022). Post hoc power is not informative. Genetic epidemiology, 46(7), 390-394.
Comment 2: The use of mixed-effects models is appropriate for the complex design. However, the reporting could be more complete to ensure reproducibility. Please specify the exact structure of the mixed models used (e.g., the random effects included, such as the intercept for each animal). Additionally, for analyses reporting degrees of freedom with decimals (e.g., F(1.934, 38.68)), please indicate if a specific correction method (e.g., Greenhouse-Geisser) was applied and the rationale for its use (e.g., violation of sphericity). This transparency is crucial for the reader to fully evaluate the statistical approach.
Response 2: We would like to thank the reviewer for addressing these concerns, we completely agree and have addressed both issues in section 2.7. We now report the fixed and random effects of the mixed-models (line 263-266). For the analyses that reported their degrees of freedom with decimals, there has indeed been a Greenhouse-Geisser correction due to the violation of sphericity. We included the usage of this correction method in section 2.7 (line 266-267).
Comment 3: The discussion section would benefit from a more nuanced interpretation of the negative findings. While the authors rightly mention potential compensatory mechanisms and the absence of maternal stress as mitigating factors, the language could more explicitly frame the results within the context of the study's limitations, particularly the statistical power point raised above. Acknowledging that the findings demonstrate a lack of a major effect of perinatal fluoxetine on these specific myelin proteins at these specific timepoints would provide a more precise and defensible conclusion. This would nicely set the stage for the excellent suggestions for future research that the authors already provide.
Response 3: We agree with the Reviewer that the interpretation of our non-significant findings regarding fluoxetine treatment need to be presented as carefully and precisely as possible. Our aim is to avoid implying that subtle effects are absent and instead to frame the conclusions appropriately within the boundaries of our experimental design. In response, we have revised the discussion to emphasize that the lack of observable effects of fluoxetine may be highly context dependent, and we have clarified this point in both the discussion (line 367–369) and the conclusion (line 460–461).
Moreover, as noted in our response to the first comment, we now acknowledge that our sample size could be too small to detect neurobiological changes induced by fluoxetine with a lower effect size (line 402-403 & 434–437).
Comment 4: The introduction effectively outlines the contradictory literature on SSRIs and myelination. The novelty of the current study—focusing on auditory regions and a specific SSRI (fluoxetine) at key developmental stages—could be more sharply articulated in the discussion. Please expand on how your findings help reconcile the discrepancies between previous studies. Does the absence of an effect with fluoxetine suggest that the robust effects seen with citalopram are compound-specific? Does it highlight the critical importance of brain region and timing? A more direct discussion positioning your null result within the existing conflicting literature would significantly enhance the impact of the manuscript.
Response 4: We thank the reviewer for this thoughtful and constructive comment. We agree that our discussion can be strengthened by more clearly situating our findings within the contradictory SSRI–myelination literature. In the revised manuscript, we have expanded the discussion to emphasize that while our previous RNA sequencing study showed altered myelin-related gene expression in the prefrontal cortex and amygdala (Ramsteijn et al., 2022), in the present work we did not observe corresponding effects at the protein level in these regions (Supplementary Material) or in auditory-related regions. We also highlight now that the robust myelination effects reported for citalopram (e.g., Simpson et al., 2011) may be compound-specific, reflecting pharmacological differences between fluoxetine and citalopram and we now stress the importance of regional and developmental specificity, since both our current null findings and previous work (Ramsteijn et al., 2022; Kroeze et al., 2016, 2017) indicate that SSRI effects on myelination are not uniform across brain areas or timepoints. Based on the previous suggestion of the reviewer we now also note that our study used modest sample sizes, which may have limited the power to detect subtle effects.This addition positions our null findings as refining rather than contradicting previous work, by delineating conditions under which fluoxetine exposure does not significantly alter myelin markers.
Comment 5: In the Methods section (2.6), it is stated that the mPFC and BLA results are in Supplementary Material 1. For completeness, it would be helpful to briefly summarize the key outcome of the fluoxetine treatment in these regions in the main text results (e.g., "Similarly, no significant effects of fluoxetine were observed in the mPFC or BLA; see Supplementary Material 1")
Response 5: We have adjusted the references to the supplementary material in the result section. We now include a brief summary of the (lack of) effect of fluoxetine on the investigated myelination markers in line 284-286, 314-315 and 334-335.
Round 2
Reviewer 1 Report
Comments and Suggestions for Authors
Can be accepted.
Author Response
Comment 1: Can be accepted.
Response 1: Thank you for taking the time to review our paper again.
Reviewer 2 Report
Comments and Suggestions for Authors
Thank you for your detailed responses to the comments. I appreciate the revisions made to the statistical reporting and the discussion, which have improved the manuscript's clarity.
Regarding your response to point #1 on statistical power, I would like to kindly point out a significant discrepancy in the sample size calculation cited. You state that for an effect size of d=0.4, alpha=0.05, and power=0.8, a sample size of 6.8 rats per group was determined. Based on a standard a priori power calculation for a two-sample t-test, these parameters would actually require approximately 100 animals per group (total N ~200), not 6.8. A sample size of ~7 per group would only be sufficient to detect an very large effect size (d > 1.5).
I strongly recommend re-checking the parameters used in your laboratory's previous power analysis.
Consequently, while your data robustly indicate that perinatal fluoxetine does not produce a large effect on the myelin markers measured, the study is underpowered to rule out more subtle, yet biologically important, effects. The conclusion should therefore be carefully framed to reflect this limitation.
A more precise summary might be: "Our findings demonstrate that perinatal fluoxetine exposure does not cause major, large-scale alterations in myelin protein levels within the auditory pathway. However, due to the limited sample size, our study cannot rule out more modest effects that may still hold functional significance."
Author Response
Comment 1: Thank you for your detailed responses to the comments. I appreciate the revisions made to the statistical reporting and the discussion, which have improved the manuscript's clarity.
Regarding your response to point #1 on statistical power, I would like to kindly point out a significant discrepancy in the sample size calculation cited. You state that for an effect size of d=0.4, alpha=0.05, and power=0.8, a sample size of 6.8 rats per group was determined. Based on a standard a priori power calculation for a two-sample t-test, these parameters would actually require approximately 100 animals per group (total N ~200), not 6.8. A sample size of ~7 per group would only be sufficient to detect an very large effect size (d > 1.5).
I strongly recommend re-checking the parameters used in your laboratory's previous power analysis.
Consequently, while your data robustly indicate that perinatal fluoxetine does not produce a large effect on the myelin markers measured, the study is underpowered to rule out more subtle, yet biologically important, effects. The conclusion should therefore be carefully framed to reflect this limitation.
A more precise summary might be: "Our findings demonstrate that perinatal fluoxetine exposure does not cause major, large-scale alterations in myelin protein levels within the auditory pathway. However, due to the limited sample size, our study cannot rule out more modest effects that may still hold functional significance."
Response 1: Thank you for taking the time to review our paper again and for providing a detailed explanation of the sample size considerations. We have addressed these points in our response below and through revisions to the text, thereby better substantiating and nuancing our results and improving the rigor of the work.
As mentioned in the first round of reviews, we conducted this study using surplus animals, therefore a priori calculation was not made. However, our previously conducted a priori sample size calculation we used for another study used G*Power to estimate the required sample size. Please note, these experiments were methodologically different from the current manuscript, since the previous study used 6 different time points and 2 groups (fluoxetine and control). That analysis was based on an F-test (ANOVA: repeated measures, within–between interaction) using the following parameters: effect size f = 0.4, α = 0.05, power (1–β) = 0.80, number of groups = 2, number of measurements = 6, correlation among repeated measures = 0.8, and nonsphericity correction ε = 1. This configuration resulted in an estimated sample size of approximately 6-7 animals per group. The chosen parameters (correlation = 0.8 and ε = 1) reflect a conservative assumption of moderate to high inter-measurement correlation and sphericity.
When adapting our previously used a priori sample size calculation for our current experimental design, adjusting the number of measurements from 6 to 2, we end up with a sample size of 8 per group (treatment per repeated measure). Although this estimate is lower than the amount estimated by the reviewer, we agree that our study is underpowered to find significant effects with a smaller effect size. Therefore, we have added additional nuanced interpretations of our findings throughout the manuscript (line 24, 39, 369, 409) following the suggestions of the reviewer. Our conclusion (line 463-466) now reads: “Our findings demonstrate that perinatal fluoxetine exposure does not cause major, large-scale alterations in myelin protein levels within auditory brain regions. However, due to the limited sample size, our study cannot rule out more modest effects that may still hold functional significance.” Lastly, we also adapted the title of the manuscript to reflect the more nuanced interpretation of our results. The title now reads: “Perinatal fluoxetine exposure has no major effect on myelin associated glycoprotein and myelin basic protein levels in auditory brain regions”.